# *PRKG2* Splice Site Variant in Dogo Argentino Dogs with Disproportionate Dwarfism

**DOI:** 10.3390/genes12101489

**Published:** 2021-09-24

**Authors:** Gabriela Rudd Garces, Maria Elena Turba, Myriam Muracchini, Alessia Diana, Vidhya Jagannathan, Fabio Gentilini, Tosso Leeb

**Affiliations:** 1Institute of Genetics, Vetsuisse Faculty, University of Bern, 3001 Bern, Switzerland; gabriela.ruddgarces@vetsuisse.unibe.ch (G.R.G.); vidhya.jagannathan@vetsuisse.unibe.ch (V.J.); 2Institute of Veterinary Genetics “Ing. Fernando Noel Dulout”, National University of La Plata, La Plata B1900, Argentina; 3Genefast Srl, 47100 Forlì, Italy; me.turba@genefast.com; 4Lupo Alberto Veterinary Clinic, 40038 Vergato, Italy; m.muracchini@icloud.com; 5Department of Veterinary Medical Sciences, University of Bologna, 40126 Bologna, Italy; alessia.diana@unibo.it (A.D.); fabio.gentilini@unibo.it (F.G.)

**Keywords:** *Canis lupus familiaris*, linkage analysis, homozygosity mapping, whole genome sequencing, bone, growth, development

## Abstract

Dwarfism phenotypes occur in many species and may be caused by genetic or environmental factors. In this study, we investigated a family of nine Dogo Argentino dogs, in which two dogs were affected by disproportionate dwarfism. Radiographs of an affected dog revealed a decreased level of endochondral ossification in its growth plates, and a premature closure of the distal ulnar physes. The pedigree of the dogs presented evidence of monogenic autosomal recessive inheritance; combined linkage and homozygosity mapping assigned the most likely position of a potential genetic defect to 34 genome segments, totaling 125 Mb. The genome of an affected dog was sequenced and compared to 795 control genomes. The prioritization of private variants revealed a clear top candidate variant for the observed dwarfism. This variant, *PRKG2*:XM_022413533.1:c.1634+1G>T, affects the splice donor site and is therefore predicted to disrupt the function of the *PKRG2* gene encoding protein, kinase cGMP-dependent type 2, a known regulator of chondrocyte differentiation. The genotypes of the *PRKG2* variant were perfectly associated with the phenotype in the studied family of dogs. *PRKG2* loss-of-function variants were previously reported to cause disproportionate dwarfism in humans, cattle, mice, and rats. Together with the comparative data from other species, our data strongly suggest *PRKG2*:c.1634+1G>T to be a candidate causative variant for the observed dwarfism phenotype in Dogo Argentino dogs.

## 1. Introduction

Skeletal dysplasias are a large, heterogeneous group of genetic disorders characterized by the abnormal growth, development and remodeling of bones and cartilage [1]. The Nosology Committee of the International Skeletal Dysplasia Society recognizes 461 different diseases, which are classified into 42 groups based on their clinical, radiographic, and molecular phenotypes. Next-generation sequencing technologies have allowed for the identification of the causal variants affecting 437 different genes in 425 of these disorders [2]. Dwarfism is the medical term used to denote short stature, and according to the Online Mendelian Inheritance in Man (https://www.omim.org/ (accessed on 16 September 2021), there are more than 200 different types of underlying skeletal dysplasias in patients with primary or secondary dwarfism phenotypes. Based on the specific phenotype, dwarfism can be classified as proportionate or disproportionate. In proportionate dwarfism, normal body proportions are conserved, while the overall height is reduced. Disproportionate dwarfism involves relative length alterations of the limbs and changes in body proportions [3,4]. The skeletal dysplasias typically present as disproportionate short stature in childhood [4].

In certain livestock animals, some dwarfism phenotypes are part of the breed standard, such as in Chinese Banna miniature pigs [5], Shetland Ponies [6], Dexter cattle [7], or West African Dwarf goats [8]. Sometimes, the dwarfs are obligate heterozygotes, and homozygous mutants are nonviable [9].

Dogs represent the mammalian species with the most extreme phenotypic skeletal diversity. Since their domestication, dogs have been subjected to human selection based on different sizes and skeletal morphologies [10]. Some alterations in limb length, which lead to disproportionate dwarfism, are a result of positive selection in some breeds. An example of disproportionate dwarfism caused by positive selection is the common chondrodysplasia caused by an *FGF4* retrogene insertion on chromosome 18 in the Dachshund, Basset Hound, and many other short-legged breeds [11]. However, there are also forms of disproportionate dwarfism that are undesired, or that are associated with health problems and represent animal welfare problems. For instance, another *FGF4* retrogene insertion on chromosome 12 causes chondrodystrophy, which is associated with the abnormal calcification of intervertebral discs and a hugely increased risk for disk prolapses [12] (OMIA 000157-9615). A rare form of chondrodysplasia is caused by a nonsense variant in the *ITGA10* gene present in Karelian bear dogs and Norwegian Elkhounds [13] (OMIA 001886-9615). Skeletal dysplasia 2 (SD2) is a mild form of disproportionate dwarfism in Labrador Retrievers, and the underlying causative variant is a missense variant in *COL11A2* [14] (OMIA 001772-9615). Osteochondrodysplasia is produced by a 130 kb deletion in the *SLC13A1* gene in Miniature Poodles [15] (OMIA 001315-9615). Spondylocostal dysostosis is characterized by truncal shortening, extensive hemivertebrae, and rib anomalies in Miniature Schnauzers and is caused by a frameshift variant in the *HES7* gene [16] (OMIA 001944-9615).

The present study was initiated after a breeder reported on Dogo Argentino dogs with disproportionate dwarfism. The goal of our study was to characterize the phenotype and identify a possible underlying causative genetic defect for such a case.

## 2. Materials and Methods

### 2.1. Diagnostic Imaging

Radiographic images of distal forelimbs and hip joints were acquired using a high-frequency X-ray unit (Raffaello HF/40, ACEM s.p.a, Italy) assembled with the DR system (Carestream DRX, Carestrem Health, Milano, Italy). Neutral mediolateral and craniocaudal projections of both the radius and ulna, as well as a ventrodorsal projection of the hip joints with an abduction of femurs were obtained. Radiographic images were recorded in the DICOM format and transferred to a computer. An image processing software (OsiriX Lite—Version 11, Pixmeo SARL, 2019) was used to view the images and to perform the evaluation. Total-body computed tomography (CT) was performed under general anesthesia using a 64-slice CT scanner (Brilliance CT 64 channel Philips Medical Systems Nederland) on patients in sternal recumbency. The acquisition parameters and the filter algorithm were adjusted in accordance with the different body regions. CT images were transferred to a workstation using a commercially available DICOM image viewing software (OsiriX Lite—Version 11, Pixmeo SARL, 2019) for the evaluation. For CT of the forelimbs and hindlimbs, multiplanar reconstruction was performed to obtain transverse, sagittal, and dorsal reformatting images.

### 2.2. DNA Isolation

Genomic DNA was isolated from EDTA blood with the Maxwell RSC Whole Blood Kit using a Maxwell RSC instrument (Promega, Dübendorf, Switzerland).

### 2.3. Linkage and Homozygosity Mapping

The Dogo Argentino family, consisting of three unaffected parents, three unaffected grandparents, two affected offspring, and one unaffected offspring, were genotyped on the Illumina canine_HD BeadChip containing 220,853 markers (Neogen, Lincoln, NE, USA). The raw SNV genotypes are available in Appendix A. The genotype data were used for a parametric linkage analysis. By using PLINK v 1.09 [17], markers that were located on the sex chromosomes, or missing in any of the nine dogs, that contained Mendel errors or had a minor allele frequency < 0.01 were excluded. For the parametric linkage analysis, an autosomal recessive inheritance model with full penetrance, a disease allele frequency of 0.5, and Merlin software [18] were used. We considered all intervals with α = 1 as linked regions that might harbor potential causative genetic defects.

The genotype data of the two affected half-siblings were used for homozygosity mapping. Markers with missing genotypes were excluded. The option for a homozygous group in PLINK was used to obtain pools of overlapping and potentially matching segments. The output intervals were matched against the intervals from the linkage analysis using Excel spreadsheets in order to find overlapping regions, which were then considered critical intervals. The output of the linkage and of homozygosity mapping is given in Appendix A.

### 2.4. Whole Genome Sequencing of an Affected Dog

An Illumina TruSeq PCR-free DNA library with a ~400 bp insert size of an affected Dogo Argentino was prepared. We collected 221 million 2 × 150 bp paired-end reads on a NovaSeq 6000 instrument (25.3× coverage). The reads were mapped to the CanFam3.1 dog reference genome assembly as previously described [19]. The sequence data were deposited under study accession PRJEB16012 and sample accession SAMEA8157167 at the European Nucleotide Archive. 

### 2.5. Variant Calling

Variant calling was performed as described [19]., The SnpEff [20] software together with the NCBI annotation release 105 for the CanFam 3.1 genome reference assembly was used to predict the functional effects of the called variants. To perform the variant filtering, we used 795 control genomes. We employed a hard-filtering approach to identify variants at which the affected dog was homozygous for the alternate allele (1/1), while 795 control genomes were either homozygous for the reference allele (0/0) or had a missing genotype call (./.). The output of the variant filtering is given in Appendix A. The accession numbers of all the genome sequences used for the analysis are compiled in Appendix A.

### 2.6. Sanger Sequencing

The *PRKG2*:XM_022413533.1:c.1634+1G>T variant was genotyped using direct Sanger sequencing of PCR amplicons. A 195 bp PCR product was amplified from genomic DNA using AmpliTaqGold360Mastermix (Thermo Fisher Scientific, Waltham, MA, USA) together with the following primers: 5’-TGC TTA GGT GGG GAG CTA TG-3’ (Primer F) and 5’-AAG AAA ACA CCA AAC ACC ATC A-3’ (Primer R). PCR was performed with an initial long denaturation of 10 min at 95 °C, followed by 30 cycles of 30 s denaturation at 95 °C, 30 s annealing at 60 °C, and 60 s polymerization at 72 °C., A final extension of 7 min at 72 °C was performed at the end. A 5200 Fragment Analyzer was used for the quality control of the PCR products (Agilent, Santa Clara, CA, USA). After treatment with shrimp alkaline phosphatase and exonuclease I, PCR amplicons were sequenced on an ABI 3730 DNA Analyzer (Thermo Fisher Scientific). Sanger sequences were analyzed using the Sequencher 5.1 software (GeneCodes, Ann Arbor, MI, USA).

## 3. Results

### 3.1. Clinical Investigations and Phenotype Description

A breeder of Dogo Argentino dogs identified two half-sibling puppies, one male and one female, with skeletal deformities that became evident at 2 months of age. The dwarfism phenotype in the male puppy born in 2018 was quite severe, and required multiple surgical interventions (Figure 1).

At 3 months of age, the most evident abnormalities in the male case consisted of limb shortening with increased angular deformities manifested at the forelimbs, which were rotated outwards (*carpus valgus*) (Figure 2A). The forelimb skeletal abnormalities had already caused gait abnormalities. Due to these abnormalities, the dog was presented to an orthopedic veterinary surgeon who diagnosed a premature bilateral closure of the distal ulnar physes. The radiographic findings indicated an asynchrony of growth between the radius and ulna, causing humeroulnar incongruity. The radiograph presented an insufficient calcification at the growth plate during bone formation. To correct the condition, the dog underwent corrective surgery consisting of ulnar elongation using dynamic proximal ulnar ostectomy, which was followed a month later by hemiepiphysiodesis of the proximal and distal radius physes (Figure 2B).

At 10 months of age, the dog showed signs of hip dysplasia with lameness in the left hindlimb, and moderate muscle atrophy had, by this point, developed. Radiographic examination showed the need for severe hip joint remodeling, and so a total hip joint replacement was carried out.

In addition to the orthopedic appendicular abnormalities, other signs of disproportionate dwarfism became evident during adolescence. At 2 years of age, the phenotype was characterized by short legs and a proportionally shorter body and neck. The head had developed a relatively broad face, a slightly upward-turned nose and a well-demarcated stop. A pronounced vertical hollow groove between the eyes was also evident (Figure 1C). The altered head and face morphology was also reflected in its altered skull dimensions. The distance between the most lateral point of the zygomatic arch was greater than for the dog’s breed standard (Figure 2C, Appendix A).

The affected female half-sibling was also born in 2018. Her phenotype was ascertained using clinical photographs (Appendix A), because the dog was not available for a detailed clinical and radiological examination.

### 3.2. Genetic Analysis

The two half-siblings with disproportionate dwarfism were from a highly inbred family. The affected puppies were born of normal parents. The pedigree relationships suggest a monogenic autosomal recessive mode of inheritance of the trait (Figure 3). A linkage analysis of this family revealed 48 linked genome segments, comprising 711 Mb in total. Subsequently, we applied a homozygosity mapping approach to the two affected half-siblings. They shared 78 homozygous regions across the genome. The intersection of the linked and homozygous intervals consisted of 34 genome segments totaling 125 Mb, corresponding to roughly 5% of the 2.4 Gb dog genome (Appendix A).

To obtain a comprehensive overview of all the variants at the critical interval, we sequenced the whole genome of an affected dog at 25.3× coverage. SNVs and short indels were called with respect to the CanFam3.1 reference genome assembly. We then compared these variants to the whole genome sequencing data of 9 wolves and 786 control dogs from genetically diverse breeds (Table 1).

Only three private protein-changing variants were identified in the linked and the homozygous regions (Table 2). A prioritization of these variants according to functional knowledge of the affected genes revealed a clear top candidate variant for the observed disproportionate dwarfism: *PRKG2*:XM_022413533.1:c.1634+1G>T. The other two variants affected genes that are not known to be involved in skeletal development or growth.

The *PRKG2* gene encodes the protein kinase cGMP-dependent type 2. The *PRKG2* variant in the affected dogs represented a splice donor variant predicted to completely abrogate the *PRKG2* function. We confirmed this variant by Sanger sequencing (Figure 4) and determined the genotypes in all the dogs of the studied family. Both cases were homozygous for the mutant allele, while the parents, one grandparent, and the unaffected half-sibling were heterozygous carriers. The other unaffected relatives did not carry the mutant allele (Figure 3).

## 4. Discussion

In this study, we identified a new form of disproportionate dwarfism in Dogo Argentino dogs. The introduction of this Argentinean dog breed to Europe happened around 50 years ago and there are only a few kennels they can be found at in Europe today [21]. This may have contributed to a particularly high level of inbreeding in the European population and may have promoted the emergence of a new recessive disease.

Using a positional candidate gene approach, along with whole genome sequencing, we identified the *PRKG2*:c.1634+1G>T splice site variant as the most likely causative variant for inherited disproportionate dwarfism.

In jawed vertebrates, there are two forms of the cGMP-dependent protein kinase: types 1 and 2, which are encoded by distinct genes, *PRKG1* and *PRKG2*, respectively. In humans, *PRKG2* is highly expressed in the bones (osteoblasts and chondroblasts), the intestines, the brain, and the kidneys [22]. It represents a good functional candidate gene for the disproportionate dwarfism phenotype due to its role in growth plate organization. Bone formation occurs through two different mechanisms: membranous and endochondral ossification. Most of the craniofacial bones develop through membranous ossification. In contrast, the longitudinal growth of long bones and vertebrae is achieved through the process of endochondral ossification in the cartilaginous growth plate, which results from the proliferation and hypertrophy of chondrocytes, and from cartilage matrix synthesis [23]. At the growth plate, PRKG2 is expressed predominantly in late proliferative and pre-hypertrophic chondrocytes and promotes their hypertrophic differentiation [24].

PRKG2 signaling leads to the phosphorylation of the SRY-box transcription factor 9 (SOX9) and the inhibition of its nuclear entry. SOX9, a member of the SOX family, is a potent inhibitor for the hypertrophic differentiation of chondrocytes, and thereby regulates chondrogenesis [25]. Unphosphorylated SOX9 activates the expression of collagen type II, a major cartilage matrix protein. Following the phosphorylation of SOX9, the collagen expression switches from collagen type II of the proliferative state to collagen type X in the hypertrophic state [26]. Previous immunohistochemical studies have shown a lack of SOX9 localization in the hypertrophic zone of the wild-type growth plate, whereas nuclear localization of SOX9 was visible in the intermediate layer of the abnormal growth plate [27]. Hence, PRKG2 has a role to play as a molecular switch, coupling the cessation of chondrocyte proliferation and the start of the hypertrophic and differentiation growth phases [26].

In humans, homozygous *PRKG2* loss-of-function variants were identified in two girls with severe disproportionate short stature due to various factors, including acromesomelic limb shortening, brachydactyly, mild to moderate platyspondyly, and progressively increasing metaphyseal alterations of the long bones. The parents were heterozygous carriers in each case. These findings were also supported by functional experiments [28].

Disproportionate dwarfism in American Angus cattle is caused by a nonsense variant in *PRKG2* (OMIA: 001485-9913). The transfection of human hepatoma cells with the mutant bovine allele reduced the ability of PRKG2 to inhibit SOX9-mediated downregulation of the collagen type II expression [29].

*Prkg2* homozygous null mice were reported to exhibit dwarfism with abnormal skull morphology and short limbs and vertebrae, caused by a severe defect in endochondral ossification at the growth plates [30,31]. The Komeda miniature rat Ishikawa is a naturally occurring mutant caused by an autosomal recessive mutation involving a 220 bp deletion that lacks the entire kinase domain of *Prkg2*. These rats showed an expanded growth plate and impaired bone healing with abnormal accumulation of non-hypertrophic chondrocytes that led to longitudinal bone growth retardation [27].

The *PRKG2*:c.1634+1G>T variant identified in Dogo Argentino dogs most likely leads to a complete loss of function of the gene. Unfortunately, we could not perform experimental follow-up analyses on the transcript or protein level of the dogs to provide functional proof for this hypothesis, as no suitable tissue samples from the affected and control dogs were available. However, the genotypes at the splice site variant co-segregated with the disproportionate dwarfism in the studied family, and the mutant allele was absent from 795 genomes of genetically diverse dogs and wolves. These data, taken together with the knowledge on the functional effect of *PRKG2* variants in humans, cattle, mice, and rats with disproportionate dwarfism phenotypes, strongly suggest that it is indeed the causative variant for the observed dwarfism phenotype in Dogo Argentino dogs.

The disproportionate dwarfism phenotype affected limb conformation, body length and height, as well as skull morphology. Morphometric analyses of the limbs were not performed in light of the intervening corrective surgery previously performed. Instead, an evaluation of the neurocranium and the viscerocranium morphology was carried out using accurate 3D-CT reconstruction. A precise, systematic and, ultimately, ontological collection of the phenes (deep phenotyping) becomes increasingly important as the number of known hereditary disorders very rapidly increases, along with an awareness that different genetic variants are responsible for confounding phenotypes. While this has already been well established in human medicine, the characterization of atypical clinical cases in animals is also of pivotal importance to veterinary medicine and science [32].

Our results allow for the introduction of genetic testing, and a breeding program that will facilitate the diagnosis and subsequent eradication of this disproportionate dwarfism from the Dogo Argentino dog population. We recommend that future matings are planned with at least one of the animals being clear to avoid the birth of further homozygous mutant offspring. Such a breeding strategy will simultaneously facilitate the gradual reduction in the mutant allele and preserve the genetic diversity in the breed.

## 5. Conclusions

In summary, we describe a new form of inherited disproportionate dwarfism in Dogo Argentino dogs and provide the *PRKG2*:c.1634+1G>T splice donor variant as a candidate causative variant. The phenotype is inherited as an autosomal recessive trait and shows parallels with previously reported forms of disproportionate dwarfism in human patients and American Angus cattle. Our data facilitate the genetic testing of Dogo Argentino dogs to prevent the non-intentional breeding of further affected puppies.

## Figures and Tables

**Figure 1 genes-12-01489-f001:**
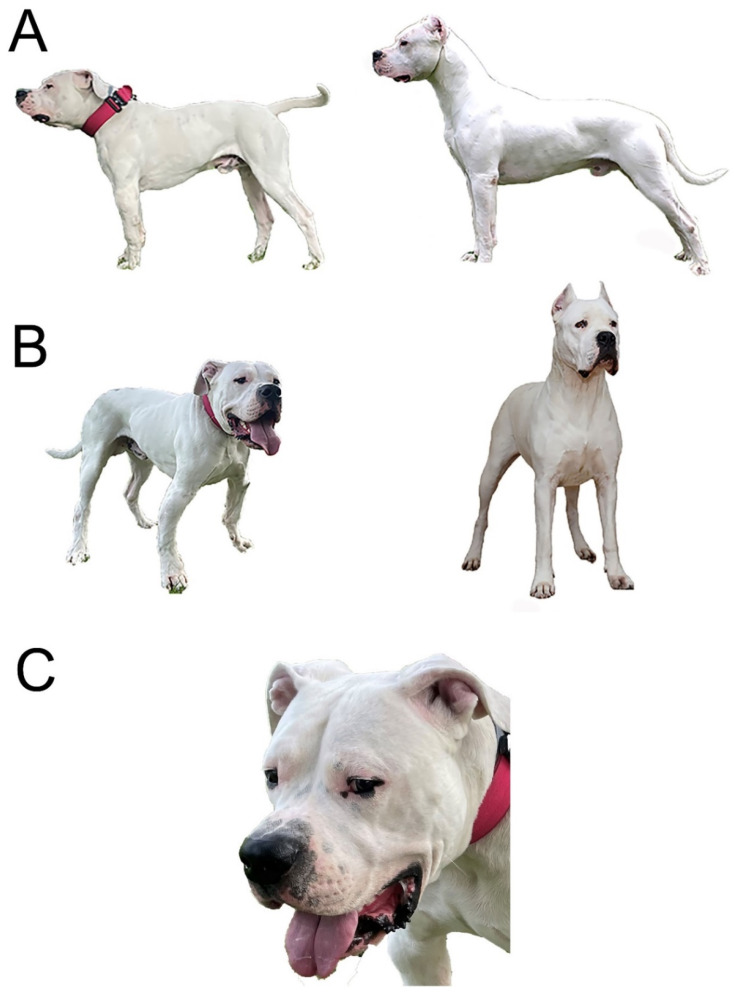
Morphology of the male affected dog at 2 years of age. The case subject (on the left) and a typical Dogo Argentino (on the right) according to the Federation Cynologique Internationale (FCI) standard. Please note that the affected dog had already undergone surgical corrections of the forelimbs before the photos were taken. (**A**) Side view: The height at withers of the case subject is 58.5 cm compared to the breed standard of 60–68 cm (ideal height 64–65 cm). The body length in the affected dog is also shorter. (**B**) View from a 45-degree angle: Disproportionally large head and forelimb angular abnormalities are evident. (**C**) Pronounced vertical groove between the eyes in the affected dog.

**Figure 2 genes-12-01489-f002:**
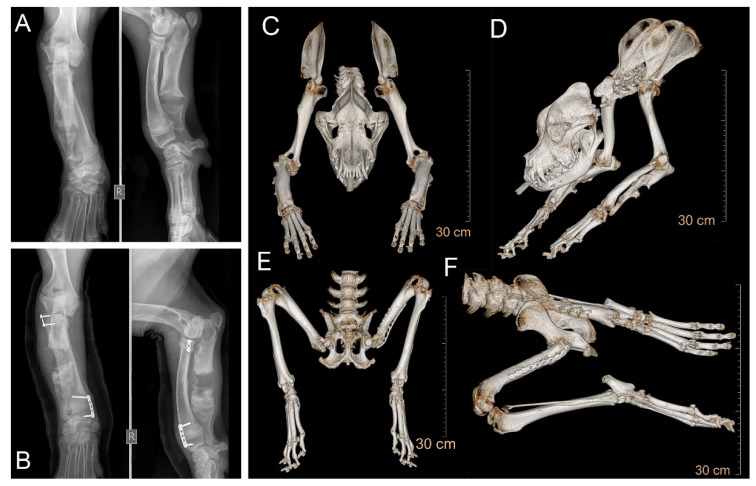
Radiological findings in the affected male dog. (**A**) Orthogonal radiographs of the frontal (left) and sagittal plane (right) of the right forelimb at 3 months of age. Angular deformities represented by shortening of ulna and subsequent bending of the radio causing incongruity of the elbow joint and carpus valgus. (**B**) Radiographs taken after surgical correction at 4 months of age. Evidence of the corrective surgery including ulnar ostectomy in mid diaphysis and presence of plates and screws to close the growth plates of the radius. (**C**–**F**) Total-body computed tomography at 2 years of age. Three-dimensional multiplanar reconstruction images highlighting the morphology of (**C**,**D**) skull and forelimbs, and (**E**,**F**) hindlimbs.

**Figure 3 genes-12-01489-f003:**
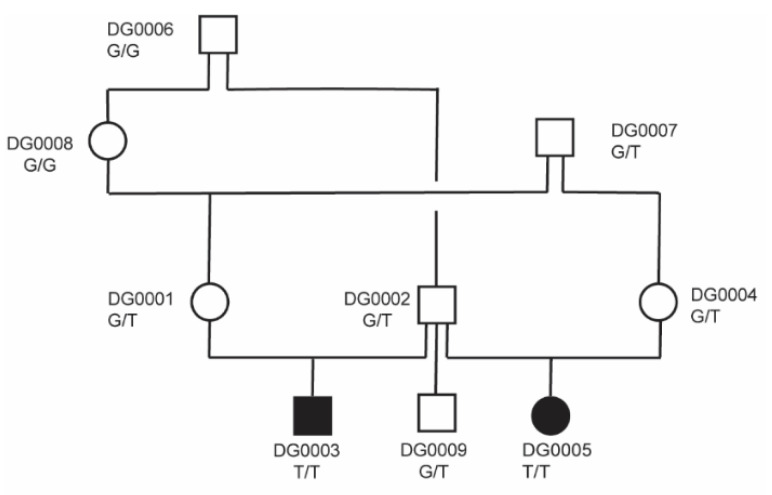
Pedigree of the investigated Dogo Argentino family. Filled and open symbols represent affected and unaffected dogs, respectively. Squares represent males, and circles females. Laboratory identifiers and genotypes at the *PRKG2*:c.1634+1G>T variant are indicated for all dogs.

**Figure 4 genes-12-01489-f004:**
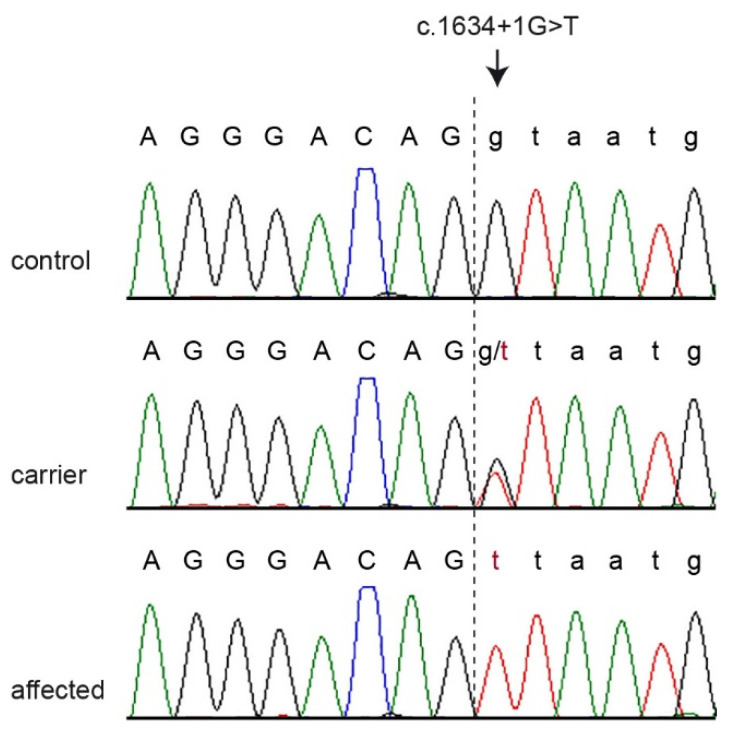
Representative Sanger electropherograms of a control, a carrier, and an affected dog illustrate the *PRKG2*:c.1634+1G>T variant. The vertical dashed line indicates the exon/intron boundary.

**Table 1 genes-12-01489-t001:** Variants detected by whole genome re-sequencing of an affected dog.

Filtering Step	Variants
Homozygous variants in whole genome	2,625,704
Private homozygous variants (absent from 795 control genomes)	2007
Private homozygous variants in 125 Mb critical intervals	196
Protein-changing private variants in critical intervals	3

**Table 2 genes-12-01489-t002:** Details of three private protein-changing candidate variants.

Chr.	Position	Ref.	Alt.	Gene	HGVS-c	HGVS-p
12	1,411,804	C	A	*NELFE*	c.7G>T	p.Val3Leu
23	50,457,119	G	C	*LEKR1*	c.206G>C	p.Arg69Thr
32	5,299,068	C	A	*PRKG2*	c.1634+1G>T	

## Data Availability

The genome sequence data used in this study are available from the European Nucleotide Archive. Accessions are given in Appendix A.

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
