# Peer review of "PRKG2 Splice Site Variant in Dogo Argentino Dogs with Disproportionate Dwarfism"

_genes, 2021, doi:10.3390/genes12101489_

Round 1

Reviewer 1 Report

The article entitled PRKG2 Splice Site Variant in Dogo Argentino Dogs with Disproportionate Dwarfism describes the identification of a mutation potentially responsible for the formation of disproportionate dwarfism in dogs. The methodological approach seems to be appropriate and the use of NGS allows for genome-wide analysis. The weak point of the manuscript is the small number of animals included in the study. Only one of the animals has full documentation of the skeletal dysfunction and the whole number of affected animals and healthy controls is only 9.  In my opinion, before this type of mutation is used in animal screening in breeding, an analysis of more animals should be carried out. In addition, before we consider the identified mutations as causal, functional studies should be performed at the level of protein and mRNA expression. Therefore, I propose to discuss these limitations and to indicate further research directions. More detailed comments:
Abstract - please indicate the number of animals used in the study.
Introduction - Please provide information about the scale of the problem - how often disproportionate dwarfism appears in dogs and Dogo Argentino Dogs specifically
Table 3 What does Hgvsc and Hgvsp mean?
line 183 How was the phenotype of the second affected dog reported? Could you provide the photo?

Author Response

(1)

The article entitled PRKG2 Splice Site Variant in Dogo Argentino Dogs with Disproportionate Dwarfism describes the identification of a mutation potentially responsible for the formation of disproportionate dwarfism in dogs. The methodological approach seems to be appropriate and the use of NGS allows for genome-wide analysis. The weak point of the manuscript is the small number of animals included in the study. Only one of the animals has full documentation of the skeletal dysfunction and the whole number of affected animals and healthy controls is only 9. In my opinion, before this type of mutation is used in animal screening in breeding, an analysis of more animals should be carried out. In addition, before we consider the identified mutations as causal, functional studies should be performed at the level of protein and mRNA expression. Therefore, I propose to discuss these limitations and to indicate further research directions.

Response: We agree with the reviewer that the study involved only a limited number of animals. We respectfully disagree with the opinion of the reviewer that this is insufficient to recommend the introduction of genetic testing. While we admittedly did not confirm the pathogenicity of the reported variant by functional experiments (discussion, lines 276-279), we still think that there is enough evidence to claim causality for the reported variant. When we follow the standards of human medical genetics (ACMG/AMP guidelines; Richards et al. Genet Med 2015, 17:405-424), we have 1 very strong evidence of pathogenicity (PVS1) with a null variant (splice site variant) in a gene, where loss of function is a known mechanism of disease. We additionally have a supporting evidence of pathogenicity (PP1) with the demonstrated co-segregation in multiple affected members of a family and we finally have as further lines of evidence the linkage and homozygosity mapping that excluded 95% of the genome and the fact that the reported splice site variant was absent from almost 800 genetically diverse dogs from different breeds (similar to the PM2 criterion in Richards et al. 2015). We rate the last two arguments as one additional supporting evidence for pathogenicity according to the ACMG/AMP guidelines. Thus, we have a total of 1 very strong and 2 supporting evidences of pathogenicity, which is sufficient to claim pathogenicity even in a diagnostic setting.

While we fully agree with the reviewer that caution must be employed before breeding programs based on the genetic testing for a specific genetic variant are recommended, we are convinced that in this case the recommendation is warranted. This disease is rare but potentially very distressing for the affected animals (and their owners). We consider it unethical to not recommend genetic testing for this variant. Our results offer a very realistic possibility to avoid the breeding of further puppies with this defect.

(2)

More detailed comments:

Abstract - please indicate the number of animals used in the study.

Response: We added the number of Dogo Argentino dogs to the abstract (9 dogs).

(3)

Introduction - Please provide information about the scale of the problem - how often disproportionate dwarfism appears in dogs and Dogo Argentino Dogs specifically

Response: Unfortunately, to be best of our knowledge, no published reliable data on the prevalence of disproportionate dwarfism in the general dog population or Dogo Argentinos exist. We describe a new form of disproportionate dwarfism and assume that this is a rare disease, but a representative analysis of the prevalence was not the focus of our study.

(4)

Table 3 What does Hgvsc and Hgvsp mean?

Response: The abbreviation HGVS stands for Human Genome Variation Society. This is the committee that sets the nomenclature rules for variant descriptions (https://varnomen.hgvs.org/). HGVS nomenclature is mandatory in human genetics, but also widely accepted in veterinary genetics and endorsed by the International Society for Animal Genetics (ISAG).The abbreviations HGVS-c and HGVS-p are commonly used to indicate variant descriptions at the cDNA or protein level, respectively. We assume that most readers of Genes will be familiar with this terminology. If not, we leave it up to the discretion of the editor to change this into “variant at the cDNA level” and “variant at the protein level”.

(5)

line 183 How was the phenotype of the second affected dog reported? Could you provide the photo?

Response: We thank the reviewer for this import comment. We added a supplementary figure with clinical photos that document the phenotype of the second affected dog.

Reviewer 2 Report

Well written paper with great images to illustrate the phenotype in question. I have only a few minor comments and have listed them below. 

Line 42 - the term "disproportionate" in relation to dwarfism is explained in line 46. I suggest moving this sentence to the end of the paragraph, after the term has been described.  

Line 61 - suggest changing "health problems" to "health disorders" to avoid repeating the word "problems" in that sentence.

Line 88 - should it be "body regions" (plural)?

Line 101 - please clarify what defined a "non-informative" marker in this study.

Line 131 - please include information about the PCR protocol used (e.g. number of cycles, annealing temperature)

Line 136 - please correct "PRC" to "PCR"

Line 153 - please change "manifest at" to "manifested in" and add a comma before "which"

Line 160 - remove the word "to" - sentence should read "underwent corrective surgery consisting of ulnar elongation..."

Line 161-162 - shouldn't it be "proximal and distal radial physes"?

Line 173 - suggest rewording to "signs of hip dysplasia with lameness of the left hindlimb and moderate muscle atrophy developed."

Line 174 - remove the letter "a" --> "radiographic examination showed severe hip joint remodeling"

Line 189 - "suggestive of a monogenic..."

Line189 - remove the full stop inserted before the parenthesis of Figure 3.

Line 216 - heterozygous carriers - an R is missing

Line 279 - "responsible for confounding phenotypes"

Lines 282-285 - this paragraph seems a bit lost at this point of the discussion. I suggest moving it into the beginning of the discussion or even to the introduction.

Supplementary Material

Table S1 - The Table S1 file contains 3 spreadsheets with different tables in each and all of the tables are labeled as "Table S2". Please revise the file. I also noticed that three marker IDs are missing in the first table. If there is a reason for this, please clarify in text. 

Author Response

(1)

Line 42 - the term "disproportionate" in relation to dwarfism is explained in line 46. I suggest moving this sentence to the end of the paragraph, after the term has been described. 

Response: We moved the sentence as requested.

(2)

Line 88 - should it be "body regions" (plural)?

Response: Revised accordingly.

(3)

Line 101 - please clarify what defined a "non-informative" marker in this study.

Response: Thank you for spotting this. We actually only used the minor allele frequency filter (MAF <1%) to remove non-informative markers. We therefore deleted “non informative” from the methods text as this is redundant and confusing.

(4)

Line 131 - please include information about the PCR protocol used (e.g. number of cycles, annealing temperature)

Response: More details on the PCR protocol were added as requested.

(5)

Line 136 - please correct "PRC" to "PCR"

Response: Revised accordingly.

(6)

Line 153 - please change "manifest at" to "manifested in" and add a comma before "which"

Response: Revised accordingly.

(7)

Line 160 - remove the word "to" - sentence should read "underwent corrective surgery consisting of ulnar elongation..."

Response: Revised accordingly.

(8)

Line 161-162 - shouldn't it be "proximal and distal radial physes"?

Response: Revised accordingly.

(9)

Line 173 - suggest rewording to "signs of hip dysplasia with lameness of the left hindlimb and moderate muscle atrophy developed."

Response: Revised accordingly.

(10)

Line 174 - remove the letter "a" --> "radiographic examination showed severe hip joint remodeling"

Response: Revised accordingly.

(11)

Line 189 - "suggestive of a monogenic..."

Response: Revised accordingly.

(12)

Line189 - remove the full stop inserted before the parenthesis of Figure 3.

Response: Revised accordingly.

(13)

Line 216 - heterozygous carriers - an R is missing

Response: Revised accordingly.

(14)

Line 279 - "responsible for confounding phenotypes"

Response: Revised accordingly.

(15)

Lines 282-285 - this paragraph seems a bit lost at this point of the discussion. I suggest moving it into the beginning of the discussion or even to the introduction.

Response: We moved this paragraph to the beginning of the discussion and rephrased the beginning of the discussion accordingly.

(16)

Supplementary Material

Table S1 - The Table S1 file contains 3 spreadsheets with different tables in each and all of the tables are labeled as "Table S2". Please revise the file. I also noticed that three marker IDs are missing in the first table. If there is a reason for this, please clarify in text.

Response: We revised the incorrect headers of the three sheets in Table S1. The 3 instances without marker IDs are on chromosomes where already the very first (= most proximal) marker showed evidence of linkage and homozygosity. In these cases, the critical intervals started with the first base of the chromosomes and therefore no marker ID could be given. (For the other intervals, the critical intervals started one marker proximal to the first marker that showed evidence of linkage and homozygosity.)

We would like to thank reviewer 2 very much for spotting all the language errors and helping us to improve the presentation of the manuscript.

Round 2

Reviewer 1 Report

The authors addressed all my questions. I suggest to underlie the fact that the reported splice site variant was absent from almost 800 genetically diverse dogs from different breeds in the discussion to additionally support the causativity of identified mutation.

Author Response

The authors addressed all my questions. I suggest to underlie the fact that the reported splice site variant was absent from almost 800 genetically diverse dogs from different breeds in the discussion to additionally support the causativity of identified mutation.

Response: Thank you for this comment. We expanded the discussion accordingly (lines 280-282).